# Global Metabolites Reprogramming Induced by Spermine Contributing to Salt Tolerance in Creeping Bentgrass

**DOI:** 10.3390/ijms23094472

**Published:** 2022-04-19

**Authors:** Zhou Li, Bizhen Cheng, Wei Liu, Guangyan Feng, Junming Zhao, Liquan Zhang, Yan Peng

**Affiliations:** 1College of Grassland Science and Technology, Sichuan Agricultural University, Chengdu 611130, China; lizhou1986814@163.com (Z.L.); chengbizhengrass@163.com (B.C.); lwgrass@126.com (W.L.); feng0201@stu.sicau.edu.cn (G.F.); junmingzhao163@163.com (J.Z.); 2Key Laboratory of Forage and Endemic Crop Biology, Ministry of Education, Inner Mongolia University, 49 Xilinguole Road, Hohhot 010020, China

**Keywords:** antioxidant capacity, energy metabolism, osmotic adjustment, polyamines, metabolic pathway, ions homeostasis

## Abstract

Soil salinization has become a serious challenge to modern agriculture worldwide. The purpose of the study was to reveal salt tolerance induced by spermine (Spm) associated with alterations in water and redox homeostasis, photosynthetic performance, and global metabolites reprogramming based on analyses of physiological responses and metabolomics in creeping bentgrass (*Agrostis stolonifera*). Plants pretreated with or without 0.5 mM Spm were subjected to salt stress induced by NaCl for 25 days in controlled growth chambers. Results showed that a prolonged period of salt stress caused a great deal of sodium (Na) accumulation, water loss, photoinhibition, and oxidative damage to plants. However, exogenous application of Spm significantly improved endogenous spermidine (Spd) and Spm contents, followed by significant enhancement of osmotic adjustment (OA), photosynthesis, and antioxidant capacity in leaves under salt stress. The Spm inhibited salt-induced Na accumulation but did not affect potassium (K) content. The analysis of metabolomics demonstrated that the Spm increased intermediate metabolites of γ-aminobutyric acid (GABA) shunt (GABA, glutamic acid, and alanine) and tricarboxylic acid (TCA) cycle (aconitic acid) under salt stress. In addition, the Spm also up-regulated the accumulation of multiple amino acids (glutamine, valine, isoleucine, methionine, serine, lysine, tyrosine, phenylalanine, and tryptophan), sugars (mannose, fructose, sucrose-6-phosphate, tagatose, and cellobiose), organic acid (gallic acid), and other metabolites (glycerol) in response to salt stress. These metabolites played important roles in OA, energy metabolism, signal transduction, and antioxidant defense under salt stress. More importantly, the Spm enhanced GABA shunt and the TCA cycle for energy supply in leaves. Current findings provide new evidence about the regulatory roles of the Spm in alleviating salt damage to plants associated with global metabolites reprogramming and metabolic homeostasis.

## 1. Introduction

Plants suffer from various abiotic stresses such as drought, cold, freeze, heat, and salt stress in nature. These adverse environmental constraints decrease crop yield and the ornamental value of horticultural plants. Nowadays, soil salinization has become a global problem, since more than 900 million hectares of the global land area are affected by salinity and the trend of salinization is increasing due to improper agricultural management practices worldwide [1]. A high concentration of sodium (Na) and other salt ions in the soil creates osmotic stress that limits the water absorption capacity of roots leading to dehydration and ion toxicity in plants, which is the first phase when plants respond to salt stress. Oxidative stress occurs as a consequence of salt-induced osmotic stress and ion toxicity in cells [2]. For better adaption to salt stress, plants develop many appropriate responses or strategies including calcium signal perception, transcriptional regulation of stress-defensive genes, accumulation of compatible solutes and protective proteins, and Na and potassium (K) transportation and homeostasis [3,4]. However, different species, especially halophytes and glycophytes, develop different mechanisms of adaption to salt stress. It is a tremendous challenge to use saline land because most commercial crops are glycophytes and sensitive to high salt.

Salt tolerance is controlled by quantitative trait loci (QTLs). For most plant species, salt tolerance is often obtained through genetic selection at the cost of loss of other advantageous characteristics such as high production, better quality, and preferable ornamental value of plants. Phytohormones or plant growth regulators (PGRs) such as abscisic acid (ABA), salicylic acid (SA), jasmonic acid (JA), and polyamines (PAs) have been applied to improve salt tolerance of crops and horticultural plants for many years [5,6,7,8]. Spermine (Spm) is a tetramine of PAs containing a higher net charge as compared to triamine spermidine (Spd) and diamine putrescine (Put). The Spm plays a diverse function in stress adaptive regulation in plants including hydroxyl radical scavenging, osmotic adjustment (OA), transcriptional regulation, and defensive proteins accumulation due to its strong polycationic nature and role in signal transduction in cells [9]. It has been reported that adverse effects of salt stress in rice (*Oryza sativa*) seedlings could be alleviated by exogenous Spd application associated with increases in transcript levels of genes for proline, ABA, and dehydrins biosynthesis [10]. Exogenous Spm ameliorated salt stress through activating antioxidant defense to alleviate oxidative damage to photosynthetic apparatus in leaves of cucumber (*Cucumis sativus*) [11] and *Adiantum capillus-veneris* [12]. The activation of H^+^-ATPase and H^+^-PPase and the up-regulation of salt overly sensitive 1 (SOS1) and Na^+^/H^+^ exchanger (NHX1) in relation to Na^+^ transport could be attributed to Spm-enhanced salt tolerance in barley (*Hordeum vulgare*) and pumpkin (*Cucurbita pepo*) seedlings [13,14]. Despite these previous reports, the regulatory roles of Spm in salt tolerance at the whole metabolic level remain to be elucidated in perennial plants.

Salt stress causes osmotic stress and Na^+^ toxicity leading to metabolic disturbance. Organic compatible solutes such as soluble sugars, amino acids, organic acids, and betaine play critical roles as osmolytes, radical scavengers, and osmoprotectants when plants respond to salt stress [15]. Metabolomics provides a key methodology to analyze comprehensive metabolites and associated metabolic networks in plants under stressful environments [16]. The study of Yang et al. [17] found out that salt-tolerant wild soybean was able to accumulate more amino acids and organic acids, whereas the salt-sensitive common soybean (*Glycine max*) preferably enhanced the tricarboxylic acid (TCA) cycle to generate more energy in response to salt stress, which was a good example of metabolomics analysis for unveiling differential adaptation to salt stress. Creeping bentgrass (*Agrostis stolonifera*) is an important horticultural crop with sensitive salt tolerance, and soil salinization decreases its quality and also increases maintenance costs [18]. Metabolites profiling has been applied to reveal changes in metabolites related to drought, heat, and salt tolerance in creeping bentgrass or other plant species [19,20,21]. However, Spm-regulated particular metabolites and relevant metabolic pathways associated with salt tolerance are not well documented in plants. The current study focused on investigating the effects of Spm pretreatment on endogenous PAs, physiological responses, Na and K accumulation, and metabolites profiling, which helped to reveal the mechanism of adaptability to salt stress in perennial plants and also highlighted the function of Spm in plants exposed to the salt-stressful environment.

## 2. Results

### 2.1. Physiological Changes Affected by Spermine under Normal Condition and Salt Stress

Figure 1A showed phenotypic changes of creeping bentgrass induced by exogenous Spm application and salt stress. The Spm-treated plants showed better growth and less wilting than the untreated plants under salt stress (Figure 1A). The content was not significantly different among four treatments after 25 d of salt stress, and the Spm pretreatment significantly increased Spd and Spm content in leaves under salt stress (Figure 1B). Salt stress caused significant declines in RWC and OP in leaves, but Spm-pretreated grasses maintained 31% higher RWC and 11% lower OP than untreated grasses under salt stress, respectively (Figure 1C,D). Photosynthetic parameters including Pn, Fv/Fm, and PIABS were not affected by Spm pretreatment under normal conditions but significantly declined under salt stress (Figure 2A–C). The Spm-pretreated grasses exhibited an 84%, 13%, or 77% increase in Pn, Fv/Fm, or PIABS more than untreated grasses under salt stress, respectively (Figure 2A–C). The TAC also increased in response to salt stress (Figure 3A). The ∙OH, carbonyl content and EL significantly increased in Spm-pretreatment and untreated grasses in response to salt stress, the Spm pretreatment had no significant effect on ∙OH, carbonyl contents, or EL under normal condition (Figure 3B–D). The Spm-pretreated grasses showed a 103% increase, 19% decrease, or 25% decrease in ·OH, carbonyl content, and EL than untreated grasses under salt stress, respectively (Figure 3B–D). Na content, K content, and K/Na ratio were not affected by the Spm pretreatment under normal conditions (Figure 4A–C). The Na accumulated extensively in the leaves when Spm-pretreatment and untreated grasses suffered from salt stress; however, Spm-pretreatment grass had a 36% decrease in Na content compared to untreated grasses under salt stress (Figure 4A). There were no significant differences in K content among the four treatments (Figure 4B). The K/Na ratio declined significantly in response to salt stress, but the Spm-pretreated grasses maintained a 69% higher K/Na ratio than untreated grasses under salt stress (Figure 4C).

### 2.2. Metabolic Profiling Affected by Spermine under Normal Condition and Salt Stress

A total of 76 metabolites were identified in leaves of creeping bentgrass including 20 amino acids, 15 sugars, 24 organic acids, and 17 other metabolites (Figure 5A). The heat map of metabolites showed that these metabolites were commonly or differentially regulated by the exogenous Spm and salt stress (Figure 5A). Most of the metabolites were unaffected by the Spm under normal conditions (Figure 5B). Under salt stress, a 33% or 13% metabolite was significantly up-regulated or down-regulated by the Spm, as reflected by the SS + Spm vs. SS. A 41% or 39% metabolite significantly declined in SS vs. C or SS + Spm vs. C, respectively. A 29% or 41% increase in metabolites was observed in the SS vs. C or the SS + Spm vs. C, respectively (Figure 5B). As compared to the control, salt stress significantly increased the accumulation of amino acids, but decreased the accumulation of organic acids in both the Spm-pretreated and untreated plants (Figure 5C). Salt stress had no significant effect on the content of organic acids between Spm-pretreated plants and untreated plants (Figure 5C). Salt stress also significantly decreased sugar accumulation in the Spm-unpretreated plants but did significantly affect sugar content in the Spm-pretreated plants, indicating the effect of the Spm application on organic acids (Figure 5C). Exogenous Spm application significantly increased other metabolites accumulation in leaves under salt stress (Figure 5C).

### 2.3. Differentially Accumulated Metabolites Affected by Spermine under Normal Condition and Salt Stress

Salt stress significantly induced the accumulation of alanine, valine, isoleucine, methionine, proline, serine, lysine, tyrosine, phenylalanine, cyanoalanine, and glutamine in both Spm-pretreated and untreated plants (Figure 6A,B). The application of Spm further improved salt-induced accumulation of alanine, valine, isoleucine, methionine, serine, lysine, phenylalanine, and tyrosine, but decreased the accumulation of proline and cyanoalanine under salt stress (Figure 6A,B). The Spm application enhanced the glutamic acid accumulation under normal conditions and salt stress (Figure 6A). In addition, the Spm-pretreated plants also maintained significantly higher GABA and tryptophan contents than the untreated plants under salt stress (Figure 6A,B). For changes in different sugars, there were no significant differences in kestose, glucose-1-phosphate, lyxose, galactose, maltose, xylose, gentiobiose, maltotriose, sucrose, and lactose between the Spm-pretreated and non-pretreated plants under salt stress, indicating individual effects of the Spm application and salt stress on those sugars (Figure 7A,B). Although the change in fructose, tagatose, mannose, sucrose-6-phosphate, or cellobiose content was affected differentially by the Spm application and salt stress, the Spm-pretreated plants exhibited significantly higher the accumulation of those sugars than the non-pretreated ones in response to salt stress (Figure 7A,B). Most of the organic acids were not affected by the Spm application under salt stress (Figure 8A,B). The ”SS + Spm” treatment maintained significantly higher aconitic acid and gallic acid contents than the “SS” treatment (Figure 8A,B). On the contrary, the Spm-pretreated plants showed significantly lower lactic acid, glycolic acid, or amminovaleric acid content than Spm-unpretreated plants under salt stress (Figure 8A). Significantly higher glycerol, guanine, hydroxypregnenolone, threitol, tricetin, and methyl phosphate contents were observed in the Spm-pretreated plants as compared to the non-pretreated plants under salt stress (Figure 9A,B). The Spm application significantly decreased the accumulation of urea, nicotinoylglycine, maleimide, and ketocholesterol, but did not significantly affect other metabolites contents under salt stress (Figure 9A,B).

### 2.4. Metabolic Pathways Associated with Metabolites Affected by Spermine under Normal Condition and Salt Stress

A total of 47 metabolites (20 amino acids, 11 sugars, 13 organic acids, and 3 other metabolites) were assigned to integrative metabolic pathways involved in sugar and amino acid metabolisms, TCA cycle, and GABA shunt (Figure 10). A significant increase in the accumulation of Spm and Spd by exogenous application of Spm effectively improved TCA cycle and GABA shunt under salt stress, as demonstrated by higher accumulation of aconitate, GABA, glutamate, alanine, and glutamine contents in SS + Spm vs. SS. The Spm also accelerated the conversion of multiple sugars into amino acids under salt stress, but limited the metabolism of GABA shunt to produce urea and proline (Figure 10).

## 3. Discussion

In response to salt stress, plants first go through salinity-caused hyperosmotic and hyperionic stresses, followed by oxidative stress. These adverse effects result in physiological drought, ion toxicity, and oxidative damage to plants, thereby inhibiting normal metabolism and growth [22]. It is well-known that enhancements of OA and antioxidant capacities are pivotal physiological strategies for plants against salt-induced osmotic stress and oxidative damage [2]. In this study, a long period of salt stress significantly decreased leaf RWC and photosynthetic parameters (Pn, PIABS, and Fv/Fm) and also increased protein oxidation and damage to cell membranes (significant increases in carbonyl content and EL level), which indicated that creeping bentgrass suffered from severe salt stress damage. However, the increase in endogenous Spd and Spm content induced by exogenous Spm application not only effectively alleviated salt-induced declines in RWC and photosynthesis, but also improved OA, TAC, and ·OH scavenging ability. These findings suggested that Spd played positive roles in regulating water and redox homeostasis in creeping bentgrass under salt stress. Apart from antioxidants and OA, intracellular ions homeostasis, especially Na and K, is crucial for plants to survive salt stress [23]. Under salt stress, Na disrupts K uptake and also causes the imbalance of other ions [24]. Our current study found that salt stress led to a large amount of Na accumulation in leaves of creeping bentgrass, and salt-stressed plants with the Spm application exhibited significantly lower Na content than salt-stressed plants without the Spm application, but K content was not affected significantly by salt stress and Spm application. Similar results were also found in studies of Nahar et al. and Kamiab et al., which showed that Spm reduced Na uptake, but did not affect K content in leaves of pistachio (*Pistacia vera*) and mung bean (*Vigna radiata*) in response to salt stress [25,26]. The maintenance of lower Na accumulation and higher K/Na ratio could contribute to Spm-induced amelioration of salt tolerance in creeping bentgrass or other plant species.

Metabolic changes associated with metabolites reprogramming are important adaptive strategies when plants suffer from salt stress [27]. As one of the main metabolic regulators, diverse amino acids exhibit various functions such as OA, osmoprotection, antioxidant, and metabolic homeostasis contributing to adaptability to salt stress in plant species [28]. GABA, glutamic acid, and alanine are three core amino acids in GABA shunt [29]. It has been proven that favorable effects of the GABA shunt improve growth regulation, carbon and nitrogen balance, signal transduction, and metabolic homeostasis under salt stress and other environmental stresses [30,31,32]. Exogenously applied GABA was also shown to effectively improve salt tolerance of creeping bentgrass associated with enhanced OA, increased K/Na ratio, and maintenance of energy supply [19]. Furthermore, the GABA was able to mitigate salt stress damage by regulating amino acids and lipids metabolism in creeping bentgrass [33]. Glutamic acid played an important role in protein metabolism and biosynthesis of other essential amino acids. A recent study found that exogenous glutamic acid application effectively alleviated detrimental effects of salinity in lentils (*Lens culinaris*) by improving ion homeostasis and antioxidant defense [34]. When plants are exposed to abiotic stresses including salt stress, alanine has the potential to promote stress tolerance because it can be converted to osmoprotective and antioxidant compounds such as β-alanine betaine and homoglutathione [35,36]. Our current study indicated the Spm-induced salt tolerance in creeping bentgrass associated with enhanced GABA shunt was propitious to the maintenance of OA, energy cycle, and metabolic homeostasis under salt stress.

However, the Spm-pretreated creeping bentgrass increased the glutamine accumulation but maintained significantly lower urea and proline contents than untreated plants in response to salt stress in our current results. It is well known that glutamine was involved in the assimilation and recycling of NH_4_^+^ for nitrogenous metabolism related to salt tolerance in plants [37]. Our previous study also showed that significant improvement in GABA shunt by exogenous GABA application was inclined to support the TCA cycle for energy generation and conversion, but did not accelerate the accumulation of urea and proline in creeping bentgrass suffering from salt stress [19]. The reason why the Spm-regulated metabolic homeostasis did not improve the accumulation of urea and proline deserves investigation in our further study since the urea and proline could play important roles in salt tolerance in plants. In addition, the application of Spm not only further improved salt-induced increases in valine, isoleucine, methionine, serine, lysine, tyrosine, and phenylalanine contents but also enhanced tryptophan accumulation in leaves of creeping bentgrass under salt stress. Phenylalanine could be catalyzed by phenylalanine hydroxylase into the biosynthesis of tyrosine, and both phenylalanine and tyrosine are involved in sugar and lipids metabolism [38]. One of the main active metabolites of phenylalanine is salicylic acid (SA) which exhibits positive effects in mediating salt tolerance in various plant species [39,40,41]. In addition, the SA ameliorated salt tolerance of soybean is associated with the accumulation of valine, isoleucine, methionine, serine, and lysine [42] since these amino acids were regarded as important osmolytes in response to abiotic stress [43]. It has been reported that foliar application of tryptophan mitigated the negative effects of salinity stress including growth retardation and lipid peroxidation in onion (*Allium cepa*) plants [44]. These studies together with findings in the current study indicated that enhanced amino acids accumulation and metabolism were important adaptive strategies in relation to the Spm-modulated salt tolerance in creeping bentgrass.

Na accumulates extensively in plants under salt stress, but other nutrient elements’ uptake and distribution are limited by salt stress, resulting in metabolic deficit and growth retardation [45]. Being an important osmolyte and energy source, sugars play a protective role in regulating OP and the energy cycle when plants go through salt stress [46]. Many sugars such as glucose and sucrose also serve as signaling molecules participating in growth regulation and stress defense systems via crosstalk with other signaling pathways [47,48]. Salt stress significantly decreased sugar accumulation in Spm-untreated creeping bentgrass, but not in the Spm-treated plants in our current study. In response to salt stress, the Spm-treated creeping bentgrass maintained significantly higher mannose, fructose, sucrose-6-phosphate, tagatose, and cellobiose contents than the untreated plants. These findings indicated that the Spm-regulated salt tolerance could be closely associated with the maintenance of sugar accumulation and metabolism in creeping bentgrass. In addition to the regular roles of mannose in OA and energy metabolism, mannose could delay leaf senescence and improve redox homeostasis and growth under normal and stressful conditions [49,50,51]. An improvement in fructose accumulation and transport ameliorated salt toxicity owing to the biological functions of fructose as a prime energy and signaling molecule [52,53]. The study of Xu et al. based on transcriptome and metabolome demonstrated that salt-tolerant oat (*Avena sativa*) cultivar Baiyan 2 enhanced sugar metabolism for energy consumption and biosynthesis, whereas opposite findings were observed in salt-sensitive Baiyan 5 under salt stress [54]. Therefore, better salt tolerance in Spm-treated creeping bentgrass could be related to sugar metabolism contributing to delayed senescence and better maintenance of signal transduction, water balance, and growth under salt stress.

Synthesis and release of organic acids are key plant protective strategies against ion stress because of their beneficial properties of pH adjustment, energy production through respiration, and detoxification of ion toxicity in cells [55,56]. Citric acid, aconitic acid, α-ketobutytic acid, fumaric acid, and malic acid are the main intermediate metabolites of the TCA cycle which is a critical pathway for substance metabolism and energy conversion [57]. Numerous studies have found the maintenance of the TCA cycle in favor of stress tolerance of various plants suffering from drought, high temperature, and salinity stress [20,21,58]. An earlier study has demonstrated that the Spd-regulated salt tolerance was associated with a reduction in the inhibition of the TCA cycle in the roots of cucumber seedlings [59]. Although most organic acids contents did not show significant differences between the Spm-pretreated and untreated creeping bentgrasses under salt stress, the Spm-pretreated plants accumulated more aconitic acid and gallic acid than the untreated plants in response to salt stress. Protective role of gallic acid in the enhancement of antioxidant defense system to alleviate salt-induced oxidative damage in rice plants [60] has been reported. The gallic acid could also improve antioxidant capacity and decrease Na accumulation in the leaf and root of cress (*Lepidium sativum*) under salt stress [61]. In addition, the Spm also induced the accumulation of other metabolites such as glycerol in leaves of creeping bentgrass under salt stress. The glycerol can be produced as an osmotic stabilizer and major carbon source under salt stress [62]. The foliar application of glycerol effectively alleviated salt-induced growth retardation, the decline in photosynthetic pigments, and the Na accumulation in pistachio (*Pistacia vera*) plants [63]. Spm-regulated accumulation of aconitic acid, gallic acid, and glycerol could play major roles in energy production, antioxidant defense, and osmotic stabilizer when creeping bentgrass responded to a prolonged period of salt stress.

## 4. Materials and Methods

### 4.1. Plant Materials and Treatments

Creeping bentgrass seeds (cv. Penn A-4, and 5 g/m^2^) were sown evenly in a container (25 cm length, 15 cm width, and 10 cm height) that was filled with quartz sands. All containers were placed in growth chambers that provided 23/19 °C (day/night), 65% relative humidity, and 750 µmol·m^−2^·s^−1^ PAR. Seeds germinated in distilled water for 8 days and then the distilled water was replaced by Hoagland’s solution [64] for 20 days of cultivation. The 28-day-old seedlings were divided into two groups: one group was cultivated in Hoagland’s solution containing 0.5 mM Spm (Spm pretreatment) for 3 days and another group was cultivated in normal Hoagland’s solution without Spm (non-Spm pretreatment) for 3 days as the control. The Spm-pretreated and untreated seedlings were then subjected to salt stress or cultivated in normal Hoagland’s solution without salt stress. Four treatments were set: (1) C, control (non-Spm-pretreated seedlings were cultivated in Hoagland’s solution for 25 days); (2) C + Spm, control plus Spm (Spm-pretreated seedlings were cultivated in Hoagland’s solution for 25 days); (3) SS, salt stress (non-Spm-pretreated seedlings were cultivated in 150 mM NaCl for 3 days, 200 mM NaCl for 3 days, and 250 mM NaCl for 19 days); and (4) SS + Spm, salt stress plus Spm (Spm-pretreated seedlings were cultivated in 150 mM NaCl for 3 days, 200 mM NaCl for 3 days, and 250 mM NaCl for 19 days). All solutions were refreshed every day to avoid a change of concentration. Four biologic replicates (four containers) for each treatment were used for the analysis of physiological parameters and metabolomics. A leaves pool including more than 6 plants was used for each replicate.

### 4.2. Measurements of Physiological Parameters

Leaf relative water content (RWC) was calculated based on RWC (%) = [(FW − DW)/(TW − DW)] × 100% [65]. FW, DW, or TW meant fresh weight, dry weight, or turgid weight (TW), respectively. For the determination of osmotic potential (OP), fresh leaves were collected and immersed instantly in distilled water for 8 h. Fully hydrated leaves were frozen in liquid nitrogen for 20 min. Leaves were thawed at 4 °C and leaf saps were pressed. The 10 mL of sap was inserted into the osmometer (Wescor, Logan, UT, USA) to detect osmolality (c). The OP was calculated based on MPa = −c × 2.58 × 10^−3^ [66]. For the determination of electrolyte leakage (EL), fresh leaves (0.1 g) were immersed in 15 mL of distilled water for 12 h and initial conductivity (Cinitial) was detected by using a conductivity meter (model 32; Yellow Springs Instrument Co., Yellow Spring, OH, USA). Leaves were then autoclaved at 100 °C for 20 min, and maximum conductance (Cmax) was detected. The OP was calculated based on EL (%) = Cinitial/Cmax × 100% [67]. For photochemical efficiency (Fv/Fm) and performance index on absorption basis (PIABS), leaves were acclimated to darkness for 30 min and Fv/Fm and PIABS were detected by a Chl fluorescence system (Pocket PEA, Hansatech, Norfolk, UK). For the net photosynthetic rate (Pn), a single layer of leaves filled the chamber of a portable photosynthetic system (CIRAS-3, PP Systems, Amesbury, MA, USA) and Pn was recorded until the reading was stable. Total antioxidant capacity (TAC), hydroxyl radicals (·OH) scavenging ability, and carbonyl content were detected by using the Assay Kit that was purchased from Suzhou Comin Biotechnology, Suzhou, China according to the manufacturer’s instructions.

### 4.3. Measurements of Endogenous Polyamines and Na/K Content

Fresh leaves (0.2 g) were ground with 1 mL cold perchloric acid (5%, *v*:*v*) to obtain the homogenate that was incubated at 4 °C for 1 h. After being centrifuged at 4 °C for 30 min (10,000× *g*), the supernatant of the homogenate was used for benzoylation. The mixture containing 500 mL of supernatant, 2 mL of NaOH (2 M), and 10 mL of benzoylchlorides was incubated at 37 °C for 30 min. Saturated NaCl solution (2 mL) was added and mixed uniformly. For extracting benzoyl polyamine, 2 mL of cold diethylether was added to the mixture. The ether phase (1 mL) was removed to a new centrifuge tube, evaporated to dryness, and then re-dissolved in 1 mL of methanol for the determination of PAs by using high-performance liquid chromatography (HPLC, Agilent-1200, Agilent Technologies, Santa Clara, CA, USA). The benzoyl PAs extract (20 mL) was loaded onto a reverse-phase Tigerkin1 C18 column. The mobile phase was methanol–H_2_O (64:36, *v*:*v*) and the flow rate was 1 mL min^−1^ at 254 nm [68]. For the Na^+^ and K^+^ content, fresh leaves were dried in an oven at 70 °C until the weight remained constant. Dry tissues were ground to a fine powder. Powders (0.5 g) were then extracted in the mixture of HNO_3_ and HClO_4_ (5:1, *v*:*v*) for 30 min at 180 °C. After being digested, a digestive solution was detected by a flame atomic absorption spectrophotometer (Analytik Jena AG, Jena, Germany). The Na or K was detected at 589 nm or 766.5 nm, respectively.

### 4.4. Metabolites Extraction and Identification

For analysis of metabolomics, the comprehensive two-dimensional gas chromatography/time-of-flight mass spectrometry (GC-TOFMS, Pegasus 4D, LECO Corporation, St Joseph, MI, USA) was used to detect global metabolites. For metabolites extraction, separation, and quantification in creeping bentgrass, methods of [69,70] were used with some modifications. The assay method in detail has been reported in our previous study [19]. Briefly, fine powder of leaf sample (20 mg) was ground with 100 µL ddH_2_O and then mixed with 500 µL of aqueous methanol (methanol:formyl trichloride = 3:1). After being centrifugated at 12,000× *g* for 10 min, 300 µL of supernatant was mixed with 10 μL of 2 mg mL^−1^ ribitol solution (an internal standard) prior to desiccation in a CentriVap benchtop centrifugal concentrator (Labconco, Kansas City, MO, USA). Samples were re-dissolved in 80 µL of methoxyamine hydrochloride (15 mg mL^−1^) at 30 °C for 90 min, and the mixture was trimethylsilylated with 80 μL N-methyl-N-(trimethylsily) trifluoroacetamide containing 1% trimethylchlorosilane for 60 min at 70 °C. Treated samples were analyzed by using GC-TOFMS. Separation (1 μL extracted liquid) was achieved on a DB-5MS capillary column (30 m × 250 μm I.D., 0.25 μm film thickness; Agilent J&W Scientific, Folsom, CA, USA), and helium was used as the carrier gas at a constant flow rate of 1.0 mL min^−1^. The measurements were made with electron impact ionization (70 eV) at full scan mode (m/z 20–600), and an acquisition rate of 10 spectrum/second in the TOFMS setting was used. Metabolites were identified by using ChromaTOF software (v. 4.50.8.0, LECO, St. Joseph, MI, USA) coupled with commercially available compound libraries: NIST 2005 (PerkinElmer Inc., Waltham, MA, USA), Wiley 7.0 (John Wiley & Sons Ltd., Hoboken, NJ, USA).

### 4.5. Statistical Analysis

The General Linear Model procedure of SAS (version 9.1; SAS Institute, Cary, NC, USA) was used to determine the significance of physiological parameters. The significance of differences among treatments was tested using the least significance test with *p* ≤ 0.05.

## 5. Conclusions

A prolonged period of salt stress caused water loss, photoinhibition, and oxidative damage to creeping bentgrass plants. The significant increase in endogenous Spd and Spm contents induced by exogenous application of Spm significantly improved OA, photosynthetic performance, and antioxidant capacity, leading to water and redox homeostasis in leaves; this might be one of the potential mechanisms involved in the Spm-regulated salt tolerance in creeping bentgrass. The analysis of metabolomics demonstrated that the Spm regulated amino acid and sugar accumulation, but exhibited fewer effects on organic acids under salt stress. These metabolites played important roles in OA, energy metabolism, signal transduction, and antioxidant defense. More importantly, the Spm enhanced GABA shunt and the shift from GABA shunt to TCA cycle for energy supply in leaves under salt stress. Current findings suggested that the Spm application induced changes in several metabolites involved in physiological pathways underlying salt tolerance.

## Figures and Tables

**Figure 1 ijms-23-04472-f001:**
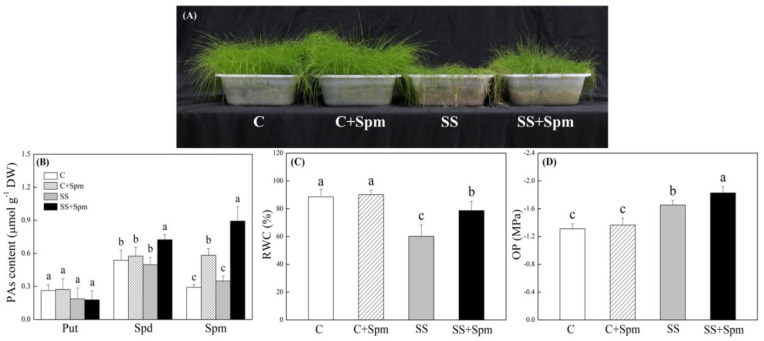
Effects of exogenous application of spermine (Spm) on (**A**) phonotypic changes, (**B**) endogenous polyamines (PAs), (**C**) relative water content (RWC), and (**D**) osmotic potential (OP) under normal conditions and salt stress. Vertical bars indicate ±SE of mean (n = 4). Different letters above columns indicate significant differences based on LSD (*p* < 0.05). C, control; C + Spm, control + Spm; SS, salt stress; SS + Spm, salt stress + Spm.

**Figure 2 ijms-23-04472-f002:**
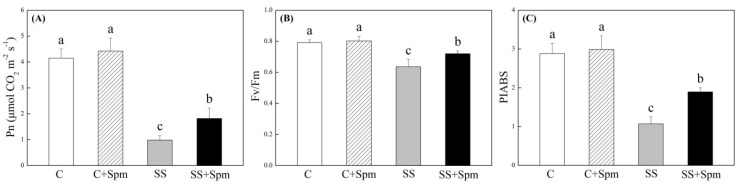
Effects of exogenous application of spermine (Spm) on (**A**) net photosynthetic rate (Pn), (**B**) photochemical efficiency of PS II (Fv/Fm), and (**C**) performance index on absorption basis (PIABS) under normal conditions and salt stress. Vertical bars indicate ±SE of mean (n = 4). Different letters above columns indicate significant differences based on LSD (*p* < 0.05). C, control; C + Spm, control + Spm; SS, salt stress; SS + Spm, salt stress + Spm.

**Figure 3 ijms-23-04472-f003:**
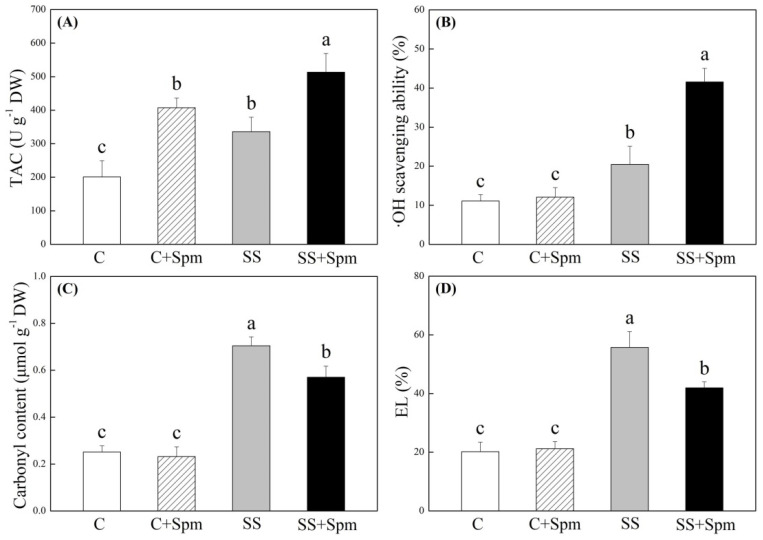
Effects of exogenous application of spermine (Spm) on (**A**) total antioxidant capacity (TAC), (**B**) hydroxyl radicals (∙OH), (**C**) carbonyl content, and (**D**) electrolyte leakage (EL) under normal conditions and salt stress. Vertical bars indicate ±SE of mean (n = 4). Different letters above columns indicate significant differences based on LSD (*p* < 0.05). C, control; C + Spm, control + Spm; SS, salt stress; SS + Spm, salt stress + Spm.

**Figure 4 ijms-23-04472-f004:**
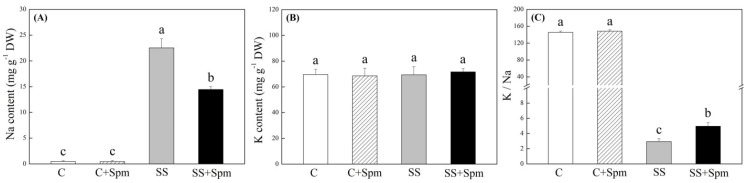
Effects of exogenous application of spermine (Spm) on (**A**) sodium (Na) content, (**B**) potassium (K) content, and (**C**) K/Na ratio under normal conditions and salt stress. Vertical bars indicate ±SE of mean (n = 4). Different letters above columns indicate significant differences based on LSD (*p* < 0.05). C, control; C + Spm, control + Spm; SS, salt stress; SS + Spm, salt stress + Spm.

**Figure 5 ijms-23-04472-f005:**
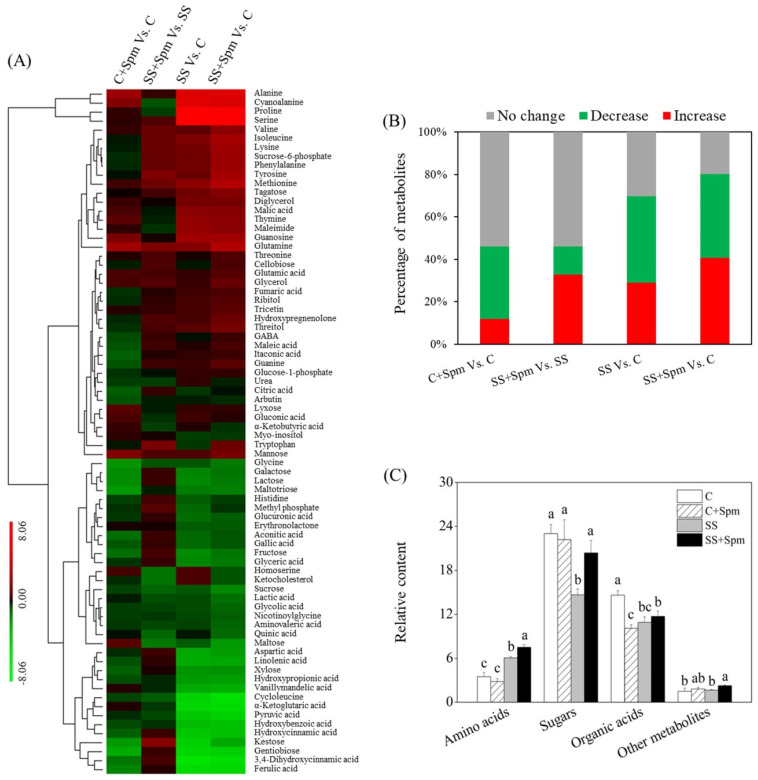
Changes in (**A**) heat map of all identified 76 metabolites, (**B**) percentage of up-regulated, down-regulated, and unchanged metabolites, and (**C**) relative content of amino acids, sugars, organic acids, and other metabolites in response to the Spm application and salt stress. Vertical bars indicate ±SE of mean (n = 4). Different letters above columns indicate significant differences based on LSD (*p* < 0.05). C, control; C + Spm, control + Spm; SS, salt stress; SS + Spm, salt stress + Spm.

**Figure 6 ijms-23-04472-f006:**
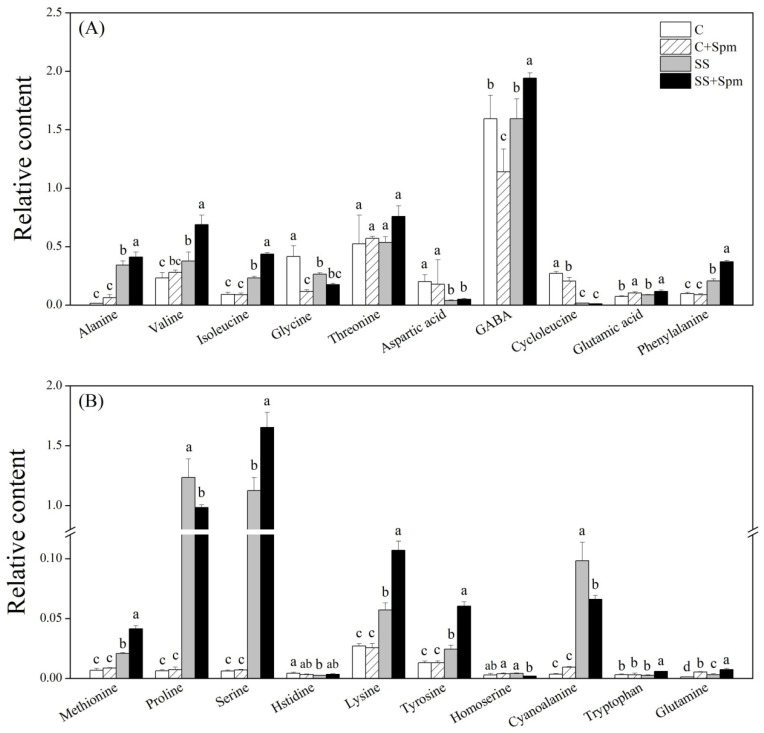
Effects of exogenous application of spermine (Spm) on (**A**) and (**B**) amino acids under normal conditions and salt stress. Vertical bars indicate ±SE of mean (n = 4). Different letters above columns indicate significant differences based on LSD (*p* < 0.05). C, control; C + Spm, control + Spm; SS, salt stress; SS + Spm, salt stress + Spm.

**Figure 7 ijms-23-04472-f007:**
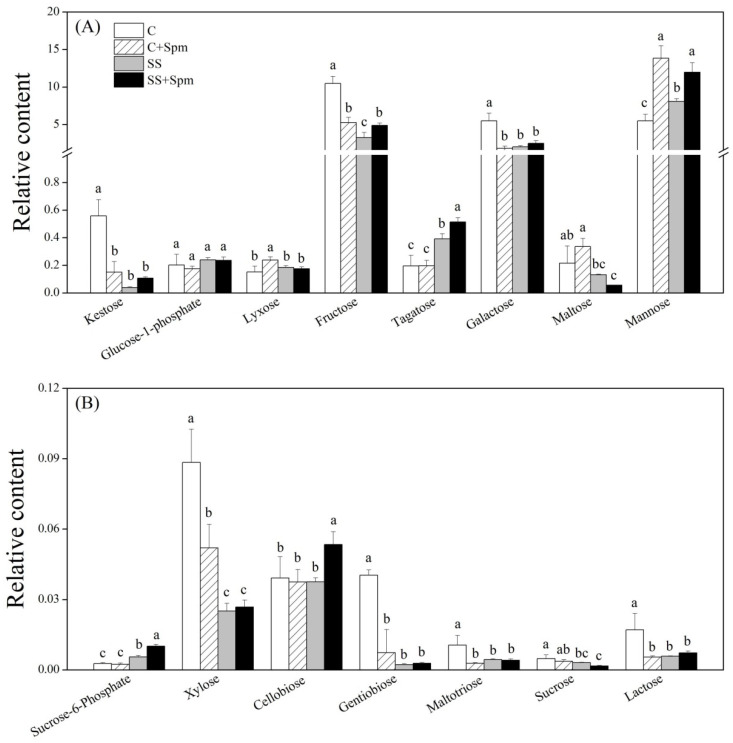
Effects of exogenous application of spermine (Spm) on (**A**) and (**B**) sugars under normal conditions and salt stress. Vertical bars indicate ±SE of mean (n = 4). Different letters above columns indicate significant differences based on LSD (*p* < 0.05). C, control; C + Spm, control + Spm; SS, salt stress; SS + Spm, salt stress + Spm.

**Figure 8 ijms-23-04472-f008:**
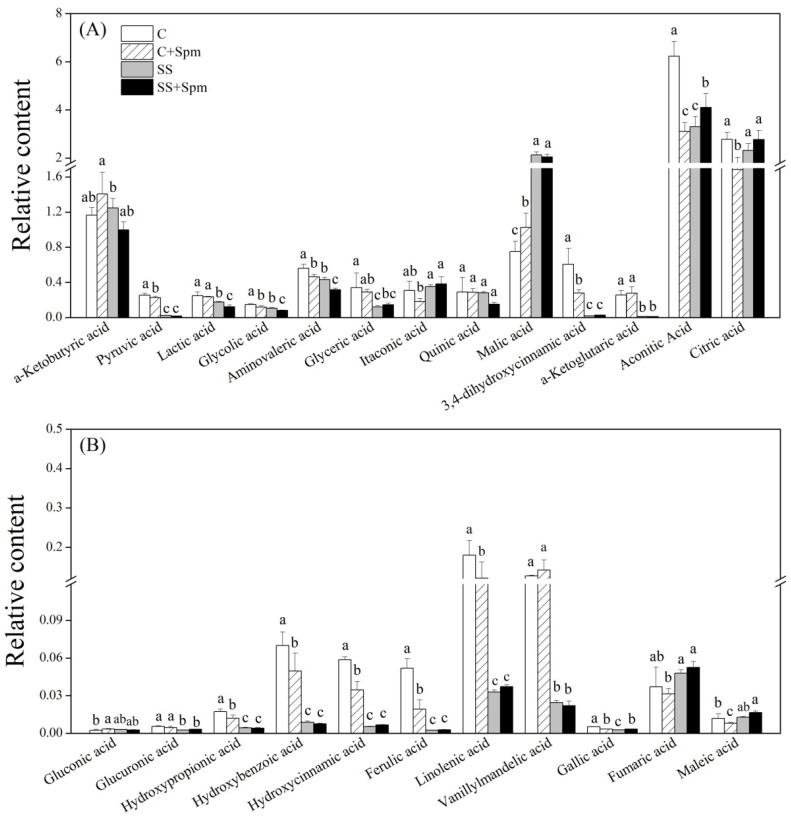
Effects of exogenous application of spermine (Spm) on (**A**) and (**B**) organic acids under normal conditions and salt stress. Vertical bars indicate ±SE of mean (n = 4). Different letters above columns indicate significant differences based on LSD (*p* < 0.05). C, control; C + Spm, control + Spm; SS, salt stress; SS + Spm, salt stress + Spm.

**Figure 9 ijms-23-04472-f009:**
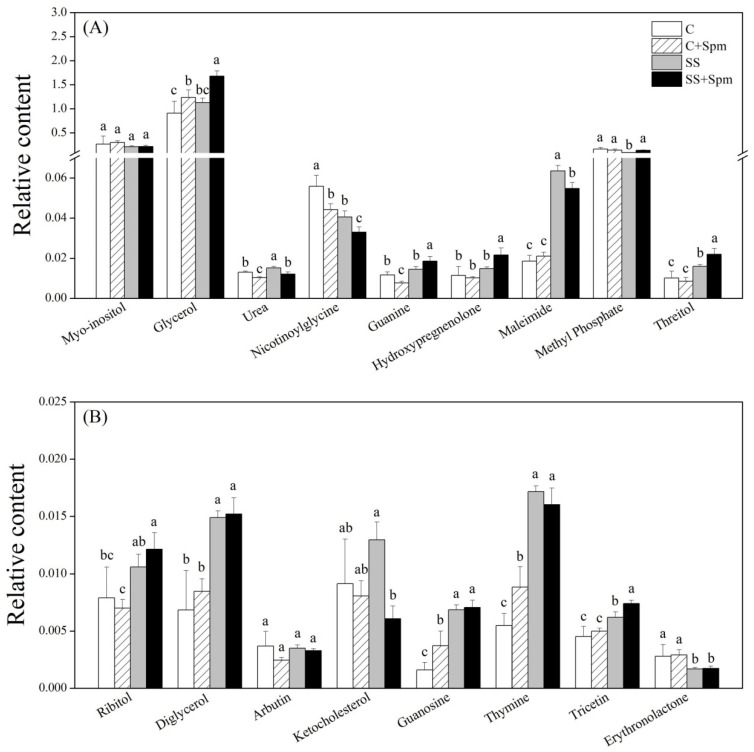
Effects of exogenous application of spermine (Spm) on (**A**) and (**B**) other metabolites under normal conditions and salt stress. Vertical bars indicate ±SE of mean (n = 4). Different letters above columns indicate significant differences based on LSD (*p* < 0.05). C, control; C + Spm, control + Spm; SS, salt stress; SS + Spm, salt stress + Spm.

**Figure 10 ijms-23-04472-f010:**
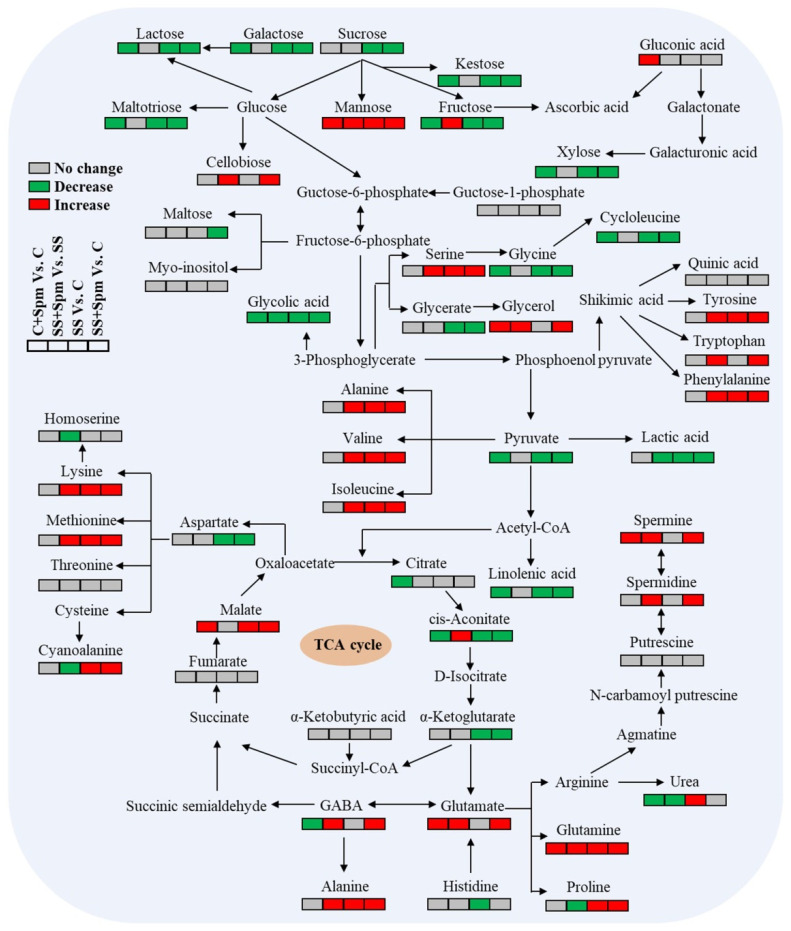
Pathway enrichment analysis of identified metabolites in leaves of creeping bentgrass in response to exogenous application of spermine under normal condition and salt stress.

## Data Availability

Not applicable.

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
