# Peer review of "Global Metabolites Reprogramming Induced by Spermine Contributing to Salt Tolerance in Creeping Bentgrass"

_ijms, 2022, doi:10.3390/ijms23094472_

Round 1

Reviewer 1 Report

The paper "Global metabolites reprogramming induced by spermine con-tributing to salt tolerance in creeping bentgrass" submitted by Li et al. is interesting. It addresses an important topic of the ability of exogeneous spermine application to mitigate salt stress in creeping bentgrass by the application of spermine, and assessment of the related metabolite changes. I recommend that the authors improve the Result presentation and Discussion. In my comments, I have among other things suggested providing data on plant growth parameters and a more consistent discussion for some of the results. I have also requested more details on methods particularly method used for metabolite data processing.  

Author Response

Comments and Suggestions for Authors

The paper "Global metabolites reprogramming induced by spermine contributing to salt tolerance in creeping bentgrass" submitted by Li et al. is interesting. It addresses an important topic of the ability of exogeneous spermine application to mitigate salt stress in creeping bentgrass by the application of spermine, and assessment of the related metabolite changes. I recommend that the authors improve the Result presentation and Discussion. In my comments, I have among other things suggested providing data on plant growth parameters and a more consistent discussion for some of the results. I have also requested more details on methods particularly method used for metabolite data processing.

Response: Thank you very much for your professional and careful review. We have revised the manuscript according to suggestions.

Suggestions marked in the PDF:

  • 1C-D. Errenous statementssalt stress significantly incresead OP. Fig. 1C. Errenous statements Spm-pretreatment further increased OP as compared to SS.

Response: Thank you for your careful review. For the change in OP, salt stress and the application of GABA significantly decreased the OP, as you can see that the Y-axis was marked by using negative numbers (Fig. 1D).

  • Add the fold change scale on Fig. 5A

Response: Thanks. The fold change scale has been added in the Fig. 5A in the left bottom corner from -8.06 to 8.06.

  • A venn diagram would be more informative. Suggestion : replace Fig. 5B by a Venn diagram.

Response: Thanks. The Venn diagram only can demonstrate numbers of metabolites, but the Fig. 5B showed the ratio of up-regulated, down-regulated, and unchanged metabolites in each group (C+Spm Vs. C, SS+Spm Vs. SS, SS Vs. C, or SS+Spm Vs. C). The Fig. 5B better showed changes in metabolites than the Venn diagram.

  • Suggestion: In Fig. 10, follow the same order of treatments as in previous figures.

Response: Thanks. Yes, we followed the same order of treatments as in previous figures. As you can see, the order of comparison groups in Fig. 5A-B and Fig. 10 was C+Spm Vs. C, SS+Spm Vs. SS, SS Vs. C, and SS+Spm Vs. C.

  • How many biological replicates of leaves were analyzed to ensure the reliability of the experimental results?

Response: Thanks. Biological replicates of leaves have been clearly described in our revised manuscript “Four biologic replicates (four containers) for each treatment were used for the analysis of physiological parameters and metabolomics. A leaves pool including more than 6 plants was used for each replicate” (Line 385-387).

  • the method used for Metabolite Data processing should be described

Response: Thank. The method used for Metabolite Data processing has been added in revised manuscript according to suggestions (Line 433-449).

  • Other suggestions and questions were marked in the PDF.

Response: Other all suggestions and questions have been revised carefully in revised manuscript according to reviewers’ suggestions.

Reviewer 2 Report

The topic of the manuscript is interesting. Authors focalized on the analysis of salt tolerance in creeping bentgrass induced by spermine associated with alterations in water and redox homeostasis, photosynthetic performance, and global metabolites reprogramming . Results showed that a prolonged period of salt stress caused a great deal of Na accumulation, water loss, photoinhibition and oxidative damage to plants. 
However, metabolomics analysis demonstrated that the spermine application can regulate the accumulation of different amino acids such as GABA and glutamic acid and sugars such as tagatose and cellobiose. These results are important if we consier the GABA role in shunt and the shift from GABA shunt to TCA cycle for energy supply in leaves under salt stress.
Generally the manuscript is well write and results well presented. Just some minor considerations:
1. in the introduction and discussion sections you should isert some more recent references. Please see the following and similar ones. Front. Plant Sci., 22 March 2022 | https://doi.org/10.3389/fpls.2022.856007; Int. J. Mol. Sci. 2022, 23(5), 2779; https://doi.org/10.3390/ijms23052779
2. in materials and metods section you must add the replicate number of samples used for analysis. You have used a leaves pool? From how many plants? 
3. Conclusion section is repetitive, the same phrases are written in the abstract. Please rephrased it.

Best regards

Author Response

Comments and Suggestions for Authors

The topic of the manuscript is interesting. Authors focalized on the analysis of salt tolerance in creeping bentgrass induced by spermine associated with alterations in water and redox homeostasis, photosynthetic performance, and global metabolites reprogramming. Results showed that a prolonged period of salt stress caused a great deal of Na accumulation, water loss, photoinhibition and oxidative damage to plants.

However, metabolomics analysis demonstrated that the spermine application can regulate the accumulation of different amino acids such as GABA and glutamic acid and sugars such as tagatose and cellobiose. These results are important if we consider the GABA role in shunt and the shift from GABA shunt to TCA cycle for energy supply in leaves under salt stress.

Response: Thank you very much for your professional and careful review. We have revised the manuscript according to suggestions.

Generally the manuscript is well write and results well presented. Just some minor considerations:

  1. in the introduction and discussion sections you should isert some more recent references. Please see the following and similar ones. Front. Plant Sci., 22 March 2022 | https://doi.org/10.3389/fpls.2022.856007; Int. J. Mol. Sci. 2022, 23(5), 2779; https://doi.org/10.3390/ijms23052779

Response: Thank you for your suggestion. These two references have been cited in revised manuscript.

  1. in materials and metods section you must add the replicate number of samples used for analysis. You have used a leaves pool? From how many plants?

Response: Yes, we used a leaves pool for each replicate in this study. Relevant explanation “A leaves pool including more than 6 plants was used for each replicate” has been added in revised manuscript (line 385-387).

  1. Conclusion section is repetitive, the same phrases are written in the abstract. Please rephrased it.

Response: Thank you for your suggestion. The conclusion section has been improved.